# Neighbourhood effects on educational attainment. What matters more: Exposure to poverty or exposure to affluence?

**Agata A. Troost**[1]*, **Maarten van Ham**[1], **David J. Manley**[1,2]

**1** Department of Urbanism, Faculty of Architecture and the Built Environment, Delft University of Technology, Delft, The Netherlands, **2** School of Geographical Sciences, University of Bristol, Bristol, United Kingdom

* a.a.troost@tudelft.nl

**Data Availability Statement:** The data that support the findings of this study are not publicly available due to privacy restrictions of Statistics Netherlands. The Microdata team of Statistics

## Abstract

Neighbourhood effects studies typically investigate the negative effects on individual outcomes of living in areas with concentrated poverty. The literature rarely pays attention to the potential beneficial effects of living in areas with concentrated affluence. This poverty paradigm might hinder our understanding of spatial context effects. Our paper uses individual geocoded data from the Netherlands to compare the effects of exposure to neighbourhood affluence and poverty on educational attainment within the same statistical models. Using bespoke neighbourhoods, we create individual neighbourhood histories which allow us to distinguish exposure effects from early childhood and adolescence. We follow an entire cohort born in 1995 and we measure their educational level in 2018. The results show that, in the Netherlands, neighbourhood affluence has a stronger effect on educational attainment than neighbourhood poverty for all the time periods studied. Additionally, interactions with parental education indicate that children with higher educated parents are not affected by neighbourhood poverty. These results highlight the need for more studies on the effects of concentrated affluence and can inspire anti-segregation policies.

## Introduction

The current interest in the economic impacts of neighbourhood effects was ignited by W.J. Wilson's book *The Truly Disadvantaged* [1]. The field has been dominated by a "poverty paradigm" ever since [2] as studies on a wide range of individual outcomes focussed almost exclusively on the presumed negative effects of living in poverty concentration neighbourhoods. The research focus on poorer neighbourhoods is understandable, as these are the places where a variety of problems accumulate and restrict individual life chances. Moreover, poor neighbourhoods are highly relevant from the perspective of public policy interventions aimed at reducing poverty and related problems. However, focusing solely on the negative effects of spatially concentrated poverty may hinder our understanding of the role of spatial context effects in individual life courses. Studying the effects of living in areas with concentrated affluence could help us to better understand how inequalities arise. After all, the Matthew effect suggests that not only do the "poor get poorer", but also that the "rich get richer" [3].

Netherlands can be reached for data access inquiries at the following e-mail address: microdata@cbs.nl. The paper also includes explanation of the Statistics Netherlands privacy agreements: https://www.cbs.nl/en-gb/about-us/organisation/privacy.

**Funding:** The research leading to these results has received funding from the European Research Council (https://erc.europa.eu/) under the European Union's Seventh Framework Programme (FP/2007-2013) / ERC Grant Agreement n. 615159 (ERC Consolidator Grant DEPRIVEDHOODS, Socio-spatial inequality, deprived neighbourhoods, and neighbourhood effects; awarded to M. v. H.), as well as from European Union's Horizon 2020 research and innovation programme (https://wayback.archive-it.org/12090/20220124075100/https://ec.europa.eu/programmes/horizon2020/) under Grant Agreement n. 727097 (RELOCAL; awarded to M. v. H.). The funders had no role in study design, data collection and analysis, decision to publish, or preparation of the manuscript.

**Competing interests:** The authors have declared that no competing interests exist.

Few studies have specifically investigated the effects of living in affluent neighbourhoods on individual outcomes [4], despite repeated calls to do so since the 1990s [5, 6]. The lack of literature on concentrated affluence is even more striking given the influential position of affluent households: the choices of the wealthy largely shape patterns of socio-economic segregation in cities, as higher income households can use their resources to select the best residential locations in a city [7]. By using their wealth, richer residents are able to [re]produce spatial inequalities, including the inequalities arising from both positive and negative neighbourhood effects [8].

To ameliorate negative neighbourhood effects, policy has often focused on the social renewal of poor neighbourhoods through relocating poor and introducing more affluent households–a policy without substantial empirical support [9]. The need to focus on tackling concentrated poverty while neglecting the spatial concentration of richer households has likely contributed to that limited policy approach [10]. Ultimately, the overwhelming focus on "fixing" poverty could, in part, be the result of researchers adopting theories based on individual social actors' attributes rather than on a more dynamic view of society, in which upper social classes manage their resources through mechanisms of exploitation and exclusion (see the overview of social inequality theories in [11]).

There is a small number of studies that have demonstrated the significant influence of elite or affluent spatial contexts on various life outcomes in Europe [4, 12–16] and in North America [17–19]. Amongst the important findings from these papers is that well-off and more highly educated neighbours can transfer their social and cultural capital through shared social networks formed within the neighbourhood. This is of particular importance for children's educational outcomes, considering that richer and more highly educated neighbours not only promote ambitious social attitudes (attending university to access high paying jobs as a norm), as well as invest in local community initiatives out of interest in the wellbeing of their own offspring [20]. Wealthier residents are likely to set higher standards for extracurricular activities for local children, spending time and resources on activities related to sport or culture. Through participating in such activities, children and teenagers not only expand their objective skills and knowledge, but also learn social codes which can be important for accessing affluent settings [21]. Evidence from the Netherlands also suggests that homogenous high-income neighbourhoods exhibit more local solidarity behaviours than poorer or mixed-income neighbourhoods [22].

This study investigates the effects of exposure to neighbourhood affluence and neighbourhood poverty on educational attainment, using data from the Netherlands. Although by international standards Dutch cities are only moderately economically segregated, there is evidence of growing socioeconomic inequality in recent years [23], as well as isolated elite spatial contexts, created by rich households seeking to further accumulate their capital [24]. Moreover, the Dutch educational system is highly stratified and shows a growing dependency on students' socioeconomic background [25]. In our study we use longitudinal register data, which enable us to follow the 1995 birth cohort and construct neighbourhood histories from birth to age 18, and measure educational outcomes at age 23. We study the effects of exposure to affluence and poverty at different stages of development: early childhood (ages 0 to 12), adolescence (13 to 17) and the entire childhood (0 to 17). The measures of neighbourhood poverty and affluence are created from bespoke neighbourhoods based on the nearest 200 households. Following earlier studies [20], we also test if the exposure to the neighbourhood context (both affluence and poverty) is different for children with different parental levels of education. We find that, in all models, neighbourhood affluence has a stronger effect on educational attainment than neighbourhood poverty. Additionally, interactions with parental education indicate that children with higher educated parents are not affected by neighbourhood poverty.

## Theoretical background

### The spatial influence of affluence

The neighbourhood context can influence educational outcomes of a child, similarly to the effect of parental and school factors, with which neighbourhood factors often interact [12]. The literature focusses mostly on social mechanisms [26] in the neighbourhood, including social interactions, which are based on physical proximity. The benefits of affluence for the quality of the built environment and facilities such as libraries, or schools, are clear–richer parents will have more resources to invest in their community, which they first carefully chose according to their preferences [27]. However, the social networks formed in the neighbourhood, which can be of high importance for children's future [4, 15], are also affected by the wealth of local inhabitants.

Much of the neighbourhood effects literature uses the theory of resource transmission through local networks, which in turn is based on Bourdieu's concepts of social and cultural capital [28]. By knowing certain types of people (social capital), individuals gain access to valuable information about schools or jobs, as well as adopt certain habits and ways of expression which lead to being accepted by those in charge of school or job admission (cultural capital). Yet even when individuals are in possession of these skills and attitudes, these paths may remain untrodden if, for example, they do not perceive attending a university as a realistic option for their future. These socially inspired possibilities are covered by the concept of habitus [29]. The life choices individuals make must fit in within their habitus, which is formed by those with whom they are interacting [30]. As individuals imitate others during their socialisation, the way they perceive the world and their place within it is shaped by their socioeconomic background. The habitus of a social class influences children's attitude to institutions [31]: the poorer parents, family members and classmates are unable to mobilise the same degree of social and cultural capital while dealing with authorities as richer ones.

Households reproduce neighbourhood characteristics by choosing neighbourhoods with people who are like themselves, and this is partly driven by their choice of housing and the neighbourhoods in which it is available [32]. Even if they are not consciously aware of social mechanisms, resourceful parents are likely to choose a neighbourhood as affluent as possible and contribute to preserving or enhancing that status [4]. Such behaviour is rationalised as a desire to provide their children with a safe environment and protect from possible disorder in other neighbourhoods rather than to seek the positive effect of affluent ones [33]. For children, a safe environment is important because they spend time with their peers outside both in early childhood and in adolescence, playing sports and games. Unsupervised play outside is less prevalent among richer children, but still present [34]. For a child from a poorer household, becoming part of a social network with children from more affluent households can result in peer effects overriding the educational and vocational preferences of their own parents [4]. Shared behaviours, such as studying together (potentially supervised or assisted by higher educated parents) or refraining from skipping class, contribute further to educational success. Parents themselves may also be affected by the parenting attitudes in the neighbourhood [20]. Neighbourhood networks are often connected to other networks, for example when local children are encouraged to join clubs playing higher status sports such as field hockey or tennis [34]. Ultimately, a transmission of resources takes place in richer neighbourhoods, and children from poorer households can benefit from residing in such places.

### Neighbourhood poverty in European context

Poorer neighbourhoods are not only deprived of resources, but also must deal with a wide range of consequences of poverty, including higher crime rates or the social isolation of

migrant groups. Many studies of neighbourhood context influencing educational attainment from the US have focused on such spatial disorder, with participants expressing the stress caused by presence of organised crime or drug trade [35, 36]. However, these issues are less prevalent in the more egalitarian European societies [37], with higher government spending on welfare [38]. There are also differences between Northern American and European urban planning, with European cities being more "urban"–denser, with well-developed public transit networks–while many American cities are characterised by extensive, car-oriented, suburbs [38]. Even if Western European cities have also experienced suburbanisation during the last decades [39], their more compact nature should result in lower spatial isolation experienced by their inhabitants. Furthermore, cities in the US have been expanding due to international migration, a phenomenon which remains much slower in Western Europe [38]. The large influx of new inhabitants from abroad may make social cohesion in American cities more difficult to achieve.

These differences between European and American cities might be a reason for caution in using US studies as inspiration for research on European data. The strong focus on poverty could be one of such trends. Even if American authors have long been calling for a greater focus on affluence [5, 6], most of the US research and public attention goes to deprived neighbourhoods [2]. Based on the practical reality of relatively egalitarian Western European cities, we assume that in the Netherlands, the lack of higher educated, affluent neighbours could be more important than the overall impact of poverty. This assumption is further supported by the few studies from European countries which show that the influence of neighbourhood affluence on various outcomes can be stronger than that of neighbourhood poverty [13, 14].

While comparing the effects of affluence and poverty, it is important to highlight that one is not simply the inverse of the other. As already discussed, poverty is often associated with crime and isolation of minority groups [35, 36]. Furthermore, the accumulation of different types of capital characteristic for affluence could progress at very different rates than the negative effects of poverty, which can also accumulate (for example, having debts can lead to difficulties in finding an affordable mortgage). There are studies which not only show that the effect of one could be stronger than the other, but also that there can be a significant effect of concentrated affluence on health while concentrated poverty has no effect at all [19]. Affluence and poverty can also interact differently with individual characteristics. This lack of symmetry is an argument for including them both in empirical models, as well as measuring them as distinct and separate factors to capture all of their influence. There are also theoretical reasons for studying poverty together with affluence, while using the Weberian-inspired conceptualisations of social and cultural capital, on which we elaborate in the next section.

### Conceptualising social inequality

This paper addresses the issue of the poverty paradigm in the literature by specifically paying attention to spatially concentrated affluence. Understanding social inequality is central in research on neighbourhood effects, as social inequality is both their cause and consequence. It is, therefore, surprising that there has been relatively little attention paid to the theorising and conceptualising social inequality itself within the field, even in the studies which do include measures of affluence. In the following sections we argue for the need of studying not only the effects of poverty, but also affluence, arising from the theories of inequality used (sometimes only implicitly) in the field.

Most of the quantitative neighbourhood effects research, including the papers discussed in the sections above, fits well into the so-called middle-range sociology, a scientific scope advocated by scientists such as Merton [40] and Boudon [41]. Middle-range sociology is situated

between the grand theories and pure empiricism, with theories focused on specific aspects of social life, instead of the whole society; it aims to identify the same social mechanisms in different situations [42]. Middle-range social research papers focus on answering specific research questions based on, most often, quantitative methods such as statistical models or experiments [43]. Studies of neighbourhood effects often investigate specific mechanisms [26], related to the effect of some form of segregation and therefore social inequality in urban space. The strict paper structure characteristic for the middle-range social studies usually does not allow for extensive theoretical commentary about inequality. Nevertheless, the concepts used in these papers are based on a variety of competing approaches to class, status and inequality (for an early overview see [44]), even if these inspirations are not immediately visible.

To understand why researchers tend to overlook the spatial effects of affluence, it is important to highlight some of the traditions in studies of social inequalities and how they relate to the neighbourhood effects field. Wright [11] outlines three main theoretical approaches within the sociology of class, social mobility and inequality: the individual-attributes approach (used in stratification research), opportunity hoarding (the Weberian approach), and mechanisms of domination and exploitation (the Marxist approach).

The individual-attributes approach focuses on how people obtain resources that allow them to attain a certain occupation, and therefore a position within the social strata. These meritocratic resources (for example, education or motivation), combined with attributes people are born with, shape their chances in life. The opportunity hoarding approach begins with the assumption that access to the most prestigious positions tends to be strongly protected–or hoarded–by those already having access. This Weberian approach studies how individuals in the higher social strata distance themselves by setting up requirements based on economic, cultural and social capital, as well as legal mechanisms of exclusion. One example, from urban geography, is when a good school is only accessible to those living in a certain district, and house prices in that area are sufficiently high that only affluent households can afford to live there. The third approach evolves around mechanisms of domination and exploitation. This Marxist approach takes the analysis further, by asserting that those who restrict access to certain resources and positions can also "control the labour of another group to its own advantage" [11]. This approach is present in urban studies research on the exploitations of tenants and ordinary homeowners by landlords and developers, and the pressure the latter can exert on government policies.

## Social inequality and neighbourhood effects

Quantitative studies on neighbourhood effects usually mix elements of the individual-attributes and opportunity hoarding approaches. The individual-attributes approach manifests itself as focus on social mobility and the idea that the position an individual ultimately attains is shaped by a bundle of attributes, many of them related to physical space. This approach has the advantage that it is relatively easy to translate into statistical models. However, because of the high level of methodological sophistication in time and space-variant predictors, researchers often reduce their most important status-related neighbourhood characteristic(s) to a single proxy variable which captures the spatial context of an individual.

One approach for measuring the affluence of a spatial context is using income [45]. Using categorical measures, or grouping neighbourhood inhabitants by their income level, often fits the research design better than using average income. Authors tend to follow the tradition of the field by focusing on poverty (choosing to create categories based on the percentage of poor households, etc.), which leads to the relatively lower number of studies on affluence [4]. From the perspective of the individual-attributes approach, this focus on poverty can be justified

because there is no assumed relationship between poverty and affluence. As such, "eliminating poverty by improving the relevant attributes of the poor—their education, cultural level, human capital—would in no way harm the affluent" [11]. By contrast, "in the case of opportunity hoarding, the rich are rich in part because the poor are poor, and the things the rich do to maintain their wealth contribute to the disadvantages faced by poor people." It therefore follows that "moves to eliminate poverty by removing the mechanisms of exclusion would potentially undermine the advantages of the affluent".

One could argue that a discussion on whether societal well-being can be improved without substantially limiting the choices or wealth of upper strata is not immediately relevant to more exploratory neighbourhood effects research. However, many neighbourhood studies still implicitly use opportunity hoarding theories to explain the mechanisms under investigation. Perhaps Maybe the most important examples are the already discussed concepts of cultural and social capital as developed by Bourdieu [29]. Bourdieu argues that social phenomena such as cultural norms are employed by upper classes to limit the access to their resources. Therefore, researching poverty in isolation disregards, potentially, the most influential part of the picture: the affluent social actors who possess the cultural, social, and economic capital. There are also theories focusing on the spread of disorder associated with capital deficiency, such as the broken windows theory [46]. It could still be illuminating to frame the commonly studied neighbourhood effects mechanisms in terms of the presence of various forms of capital, rather than a lack of it. Those studies investigating the effect of affluence often omit discussion of the wider implications of focussing on the effect of poverty in research. In addition to developing more methodologically sophisticated operationalisations of the current variables, quantitative neighbourhood effects researchers could deepen their assumptions and conclusions by grounding them in sociological theory. This is one of the goals of the current paper, although there are still interesting steps to be taken, such as questioning not only the poverty paradigm, but also the meritocracy paradigm [47] as well as expanding the conceptualisations of social class [48].

## Current study

Studies of neighbourhood effects on educational attainment (and in a broader sense all spatial effects studies) should investigate not only the effect of neighbourhood poverty, but also the effects of concentrated affluence. We argued that a better understanding of affluence is crucial for the neighbourhood effects mechanisms driven by various forms of capital. We use household income as a measure of poverty and affluence, which is highly correlated to other, more intangible, characteristics such as social cohesion [49]. Income also serves as a proxy of resources available to neighbourhood inhabitants. Using income allows us to construct detailed individual neighbourhood histories and investigate the effects of different periods of exposure. We also create bespoke neighbourhoods, which reflect local spatial ties better than neighbourhoods based on administrative borders.

Following the literature review, we expect that the positive effect of exposure to affluent neighbours on education attainment will be stronger than the negative effect of exposure to poorer neighbours. We also expect differences between the effects of exposure to contextual poverty and affluence at different developmental stages, but it is not clear from previous work which period of influence will have the greatest impact. For instance, early years childhood exposure could be more influential for educational attainment than later exposures because of values and beliefs formed during the early years. Young children also experience less disruption from changing the neighbourhood environment [50]. However, adolescents have greater freedom from their household and spend more time with their peers away from the parental control, and therefore exposures during adolescence could be more important.

In recent years the focus of neighbourhood effects research has shifted somewhat from "do neighbourhood effects exist?" to "for whom" do they matter [51]. In the case of children, social background could prevent them from interacting with poorer or richer neighbours [4]. Parents can explicitly limit children's interactions or simply not create any opportunities to play or socialise with children in other groups. On the other hand, children of higher educated parents may be more likely to believe in the importance of education regardless of their peer contacts in the neighbourhood. Given these propositions, we test for interactions between the exposure to neighbourhood affluence or poverty and parental education.

## Data & methods

For our empirical analysis we used individual level, geo-coded longitudinal register data from the Statistics Netherland's Social Statistical Database (SSD), which covers the entire population of the Netherlands. We selected 140,338 individuals born in 1995 who also had complete neighbourhood histories between 1995 and 2017, when they are around 22 years old, and without missing information on the variables of interest (except for parental education, which has a large percentage of missing values). For our dependent variable, education level, we measured the level of education attained by age 23 and translated this in the number of years someone would normally need to achieve that level. We added an extra year for those who studied at research universities (*wo*) to distinguish them from universities of applied science (*hbo*). The resulting variable ranges from the minimum of 2 years for unfinished primary education, to a maximum of 23 years required to obtain a doctoral degree, with the mean 16.5 years. For individuals who were still following education in the final year of observation, the level of education that they were following at that time is registered.

The data underlying our results cannot be shared publicly as they are a part of the confidential Statistics Netherlands data. Statistics Netherlands is legally responsible for consent related to data use and they have approved our project. CBS is bound by the European General Data Protection Regulation (GDPR). In addition, CBS adheres to the privacy stipulations in the Statistics Netherlands Act, the European Statistics Code of Practice, and its own Code of conduct [52].

### Contextual affluence and poverty

Contextual poverty is measured as a ratio and based on the Eurostat definition of the at-risk-of-poverty rate, which is the share of households with an equivalised disposable household income below 60% of the national median equivalised disposable income. The threshold for contextual affluence is set at 150% of that median, resulting in a similar percentage of the population above this threshold as the percentage of households under the poverty threshold. Even though in our data the detailed household income extends back to 2003, we have sufficient spatial information to people's residential histories all the way back to 1995, a further 8 years. To overcome the lack of neighbourhood income data pre-2003 we used the averaged neighbourhood income data from 2003 for all years between 1995 and 2002. Although neighbourhood characteristics change over time, using the 2003 data for earlier years is the only way to include the longer time period, which is crucial for our purposes (see [53] on the static nature of neighbourhood positions).

The geocoded nature of our data gives us information on the residential location for each individual at a spatial resolution of 100x100m grid squares. Using this information, we have created bespoke measures of neighbourhood affluence and poverty for each year using Equipop [54]. Equipop calculates the proportion of the *k*-nearest neighbours that meet user-set criteria, in our case a ratio of the neighbours meeting the poverty or affluence criterion within the

200 nearest households for each year of an individual's life. These ratios are the building blocks of our neighbourhood history variables, which are described in more detail below. We adjusted the income criterion for the median income in each year: households with an income above 150% of median household income that year were classified as affluent, and those with an income below 60% of median as poor. If, for example, an individual scores 0.15 for their 2005 neighbourhood affluence ratio, this means that in 2005, 15% of the 200 nearest households were regarded as affluent.

By constraining our neighbourhoods to the 200 nearest households, we are able to standardize measures both in densely and sparsely populated areas, important in this study, since we use the data from the whole country. Furthermore, as most of our predictors are based on social interaction, it is appropriate to focus on people rather than space while operationalising the variables.

The scale of spatial research should be chosen according to the theoretical assumptions of the study [55], and in our case we focus on relatively small-scale, social-interactive neighbourhood effects which would happen in neighbourhoods of about 200 households. This size should reflect a social space where people are likely to interact with each other, which, according to the assumptions of this study, assists in acquiring the skills and resources relevant for an individual's educational attainment.

### Exposure to neighbourhood affluence and poverty

We measure exposure to neighbourhood affluence and poverty by combining annual affluence/poverty ratios during different developmental periods: early childhood (ages 0 to 12), adolescence (13 to 17), and the entire childhood (0 to 17): we add up the yearly ratios and divide them by the number of years. The affluence and poverty variables in each period are only weakly correlated (correlation of -.45 for all three periods). We do not include measures of neighbourhood exposure after the age of 17; running models until the age of 23 in an earlier study has shown that young adults have very particular neighbourhood experiences. Many of them leave the parental home around the age 18, moving to cheap student accommodation in often low-income neighbourhoods. That creates a positive effect of having many poor neighbours on attained education, but as the education is rather the cause than the result in such a case, we decided to include only neighbourhood histories up to and including age 17.

### Control variables

The control variables in this study include an individual's sex (female or male) and their ethnicity, which is coded as native Dutch (both parents born in the Netherlands), Western migrant or a non-Western migrant background (Western countries, according to the Statistics Netherlands definition, are all European and Northern American countries along with Japan, Australia and Indonesia). Additionally, an individual's household context is represented by their household income measured in 2007, when the individual being observed would have been twelve years old, the age by which mothers are likely to have re-joined the labour market, and a variable recording parental education level (lower, middle, higher or missing). The latter variable is constructed by recording the highest education level achieved by either of the (up to) two parents. Parents with missing information on their education are kept in the data as a separate category because of their large number (11% missing) and an overrepresentation of migrants in this category. A control variable at the municipality level is the level of urbanicity, based on the proportion of years between 1999 and 2017 (for which the address density data was available) an individual has lived in an urban environment. To control for the density of social interactions at a lower level, we also included interval distance, measured by Equipop in

kilometres necessary to reach the 200 nearest neighbours. The descriptive statistics of all variables can be found in Table 1.

## Analytical approach

We estimated a series of linear regression models with educational level at age 23 as the dependent variable. All models are estimated on the same sample of 140,338 individuals, and contain the same control variables. Given the nested structure of our data, the use of multilevel modelling appears logical. However, there are two reasons why we have not used this type of models. Firstly, individuals are nested in neighbourhoods and these can change each year requiring multiple hierarchies which creates a complex structure inhibiting model convergence. This is further exacerbated by the second reason, whereby there is no strict hierarchy because of the multiple membership of individuals in the bespoke neighbourhoods (the neighbourhoods are overlapping with each other). Furthermore, because of bespoke neighbourhoods which are constructed for each individual every year, and only including people born in 1995 in the sample, a large number of individuals are nested alone in their neighbourhood (71,016; 50.60%), which is a further complication in estimating a hierarchical fixed effects structure.

The spatial variables contribute to around 3% difference in R-squared. The initial model without spatial variables explained around 15% (for detailed coefficients, see the S1 Appendix), increasing to 16% when the urbanicity control was added, to 18% with all spatial variables included. This is the magnitude of difference that can be expected from similar variables in sociological models. Additionally, including the spatial variables diminishes the effects of other variables in the model, such as family income, which means the spatial variables contribute to the underlying causal structures. VIF values were unproblematic, therefore there are no issues with multicollinearity in the models (see the S1 Appendix for exact VIF values).

**Table 1. Descriptive statistics (*N* = 140,338).**

| | Mean / % | SD | min | max |
|---|---|---|---|---|
| Education level (in years) | 16.482 | 1.609 | 2 | 23 |
| Exposure to neighbourhood affluence (age 0–17) | .163 | .101 | .000 | .820 |
| Exposure to neighbourhood affluence (age 0–12) | .163 | .101 | .000 | .831 |
| Exposure to neighbourhood affluence (age 13–17) | .163 | .111 | .000 | .802 |
| Exposure to neighbourhood poverty (age 0–17) | .114 | .072 | .014 | .848 |
| Exposure to neighbourhood poverty (age 0–12) | .111 | .071 | .008 | .860 |
| Exposure to neighbourhood poverty (age 13–17) | .122 | .087 | .009 | .892 |
| Female | 49% | | 0 | 1 |
| Household income (2007, in 10,000 euros) | 2.298 | 1.546 | * | * |
| Household income (in 10k euros, median centered) | .287 | 1.546 | * | * |
| Western | .052 | .221 | 0 | 1 |
| Non-Western | .133 | .341 | 0 | 1 |
| Native Dutch | .815 | .388 | 0 | 1 |
| Parental education | 1.780 | .979 | 0 | 3 |
| Lower parental education | 28% | | 0 | 1 |
| Middle parental education | 33% | | 0 | 1 |
| Higher parental education | 28% | | 0 | 1 |
| Parental education missing | 11% | | 0 | 1 |
| Urbanicity | .771 | .414 | 0 | 1 |
| Equipop distance (in km) | 0.213 | 0.282 | 0 | 7.288 |

* Removed because of Statistics Netherlands privacy regulations.

## Results

### Exposure to neighbourhood affluence and poverty

Table 2 presents the effects of exposure to neighbourhood affluence and poverty over time on educational level (measured in years) at age 23. In the case of affluence, the effects of exposure during the entire childhood (ages 0 to 17) and early years (0–12) are both positive and similar in size (b = 2.138, p < 0.001, beta = 0.133 and b = 2.119, p < 0.001, beta = 0.132, respectively). The effect of exposure to affluence during adolescence remains positive, but is smaller (b = 1.733, p <0.001, beta = 0.118). Compared to early childhood (b = -0.827, p < 0.001, beta = -0.036), the negative effect of exposure to poverty is slightly stronger when taking into account the whole childhood (b = -0.989, p < 0.001, beta = -0.043), and the effect during adolescence (b = -0.925, p < 0.001, beta = -0.052) is the strongest, when looking at the standardised beta coefficient. The most important finding for this paper is the comparison between the effects of affluence and poverty. The modelling results show that exposure to affluent neighbours has a stronger overall effect on educational attainment for all three time periods than exposure to poverty, confirming our hypothesis.

Most of the control variables have the expected effects, with women having a slightly higher levels of education level than men, and with higher parental household income and education being positively related to educational attainment. A surprising effect is that, in our models, Western and non-Western ethnic minorities have a slightly higher educational levels compared to native Dutch individuals. However, our models control both for parental household income and parental education level, which explains much of the negative influence of belonging to a minority ethnic background observed in other studies. In total, each of the models explains almost 18% of the variance in educational attainment.

**Table 2. Effects of exposure to neighbourhood affluence and poverty in childhood and adolescence on educational level at age 23 ($N$ = 140,338).**

|  | (1) |  | (2) |  | (3) |  |
| --- | --- | --- | --- | --- | --- | --- |
|  | Exposure age 0–17 |  | Exposure age 0–12 |  | Exposure age 13–17 |  |
|  | b | SE | b | SE | b | SE |
| Exposure to neighbourhood affluence | 2.138*** | (0.048) | 2.119*** | (0.047) | 1.733*** | (0.044) |
| Exposure to neighbourhood poverty | -0.989*** | (0.066) | -0.827*** | (0.066) | -0.925*** | (0.055) |
| Female | 0.309*** | (0.008) | 0.310*** | (0.008) | 0.309*** | (0.008) |
| Household income (in 10k euros, median centered) | 0.110*** | (0.003) | 0.113*** | (0.003) | 0.114*** | (0.003) |
| Western (ref. native Dutch) | 0.047** | (0.018) | 0.045* | (0.018) | 0.042* | (0.018) |
| Non-Western | 0.130*** | (0.013) | 0.119*** | (0.013) | 0.099*** | (0.013) |
| Middle parental education (ref. lower educated) | 0.437*** | (0.014) | 0.443*** | (0.014) | 0.441*** | (0.014) |
| Higher parental education | 1.258*** | (0.014) | 1.269*** | (0.014) | 1.274*** | (0.014) |
| Parental education missing | 0.677*** | (0.014) | 0.686*** | (0.014) | 0.686*** | (0.014) |
| Urbanicity | 0.325*** | (0.011) | 0.325*** | (0.011) | 0.330*** | (0.011) |
| Equipop distance | -0.250*** | (0.015) | -0.238*** | (0.015) | -0.267*** | (0.015) |
| Constant | 15.122*** | (0.019) | 15.094*** | (0.019) | 15.183*** | (0.018) |
| $R^2$ | 0.181 |  | 0.180 |  | 0.179 |  |

Standard errors in parentheses

* $p < 0.05$

** $p < 0.01$

*** $p < 0.001$

## Interactions with parental education

The effects of exposure to neighbourhood affluence and poverty remain significant in the models which include interactions between these neighbourhood factors and parental education, ranging from lower parental education (reference category), through middle, to higher education, and also including the sizable group of parents whose education level is missing from the data. In the model with interactions with neighbourhood poverty we additionally include the exposure to neighbourhood affluence as a control variable, and vice versa (for detailed results, see the S1 Appendix). For ease of interpretation, we present the results of the interaction terms visually. Fig 1 shows the slopes of the interactions from both models. In the model with the interactions with neighbourhood poverty, children from households with at least one higher educated parent do not appear to be affected by the proportion of poor households in their bespoke neighbourhood. Children of either middle or lower educated parents are negatively impacted, although the severity of the impact is differential. When the proportion of poor neighbours is low then it is the children of lowest educated who are most at risk; the experienced effects are similar for children from lower and middle educated families at the highest proportion of poor neighbours.

In the model with the interactions with neighbourhood affluence, all interaction slopes are positive, although the slope of the interaction between higher parental education and neighbourhood affluence is slightly flatter. This implies that again, children with at least one higher educated parent are less susceptible to their neighbours' influence on educational attainment, compared to those with lower educated parents. However, this difference is less pronounced in the case of exposure to affluent neighbourhoods than to poor ones.

## Conclusions & discussion

In this paper we have compared the effects of exposure to neighbourhood affluence and neighbourhood poverty during different stages of childhood on educational attainment. We argued that there are theoretical reasons to believe that exposure to affluence may actually be more important as a predictor of educational attainment than exposure to poverty, because of the crucial influence of interacting with higher educated people on one's resources, skills and educational aspirations; and, in the Dutch context, because of the lack of extreme concentrated poverty. Confirming this empirically, our results show that neighbourhood affluence has a

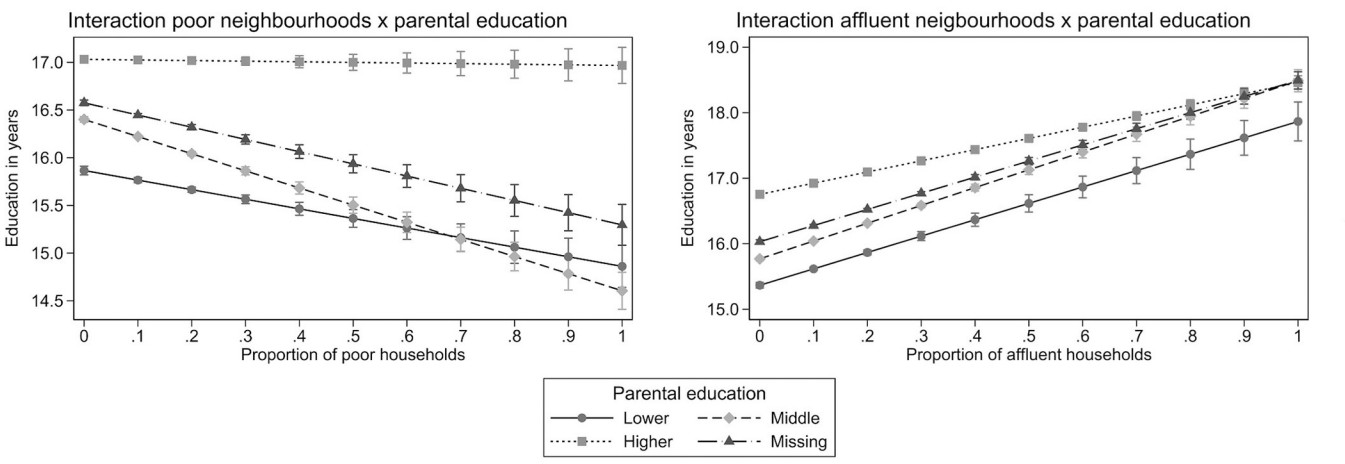

**Fig 1. Interactions between the ratio of poor or affluent neighbours and the parental education.**

stronger effect on educational attainment than neighbourhood poverty in the Netherlands. This is consistently the case across different time periods–from early childhood (ages 0–12), adolescence (13–17)–as well as for the entire childhood (0–17). According to our models the neighbourhood effects during different time periods are similar when it comes to magnitude, direction, and significance. Interestingly, the effect of exposure to poverty during the entire childhood period is stronger than that of shorter periods, which contrasts with previous results from the US [50] and the Netherlands [56].

We considered the educational level of parents to explore whether children from higher or lower educated parents are influenced differently by the neighbourhood. This is in line with earlier works, arguing that neighbourhood effects may not be the same for everybody within the neighbourhood, and that the heterogeneity of individual backgrounds might be important for their transmission [51]. The interactions between the effects of neighbourhood affluence or poverty and parental education level show that children with at least one higher educated parent are not impacted by neighbourhood poverty. We therefore consider higher education to be a buffer against negative neighbourhood contexts. However, children with higher educated parents are still influenced by neighbourhood context when that context is set in affluence, although their gains are not as great as those experience by children living in households with lower levels of parental education.

Most importantly, our results highlight how spatially concentrated affluence contributes to the reproduction of socioeconomic inequalities, as the effect of neighbourhood affluence on educational attainment is stronger than that of neighbourhood poverty. It seems that, in this sense, neighbourhood effects in the Netherlands are similar to those observed in the UK [13] and Finland [14]. Our results, specifically the effect of spatially concentrated affluence being stronger than that of poverty, support our initial idea that it is often the *lack of resources*–the cultural and economic capital of richer neighbours—in poor and middle-income neighbourhoods that is the problem, not the theorised negative effects of poverty itself. Again, in the Dutch context, crime and teenage delinquency are at relatively low levels compared to the United States, where much of the previous literature is set. Social interactions with resourceful neighbours and peers do seem to play an important role in forming children's ambitions, as well as in sharing knowledge and forming attitudes that support them. Additionally, children with at least parent with a higher level of education were less susceptible to neighbourhood influences, especially when living in poor neighbourhoods, which suggests that parental resources have a buffering role, compensating for the local lack of capital. Such children were also less affected in affluent neighbourhoods, but they still benefitted from the neighbourhood context. This implies that neighbourhood resources can have an added effect regardless of family background.

One potential possible limitation of this study is that we have measured neighbourhood resources only taking into account household income. While the use of this relatively simple variable allows for a sophisticated operationalisation of neighbourhood histories at across time periods it does not necessarily capture all important dimensions of resources. Future work could try to include other dimensions of capital and inequality to investigate the effects of living near elite, rather than just affluent, social groups. The sequences of moving from more to less affluent neighbourhoods, and vice versa, could also be studied, as we did in an earlier paper focusing on the different temporal aspects of exposure to neighbourhood poverty. Future studies should also include the role of the school context [57], with a direct measure of it. Lack of the school context is a possible limitation of this study; however, the effect of schools can be a mediating factor in the neighbourhood effect on educational achievement in the Netherlands [58]. And finally, when longer time series become available, future studies could measure educational attainment at an older age, which may provide more accurate

information on obtained diplomas and final qualifications as well as the impacts of returning to education in later adulthood.

In the introduction we observed that neighbourhood effects research is trapped in the poverty paradigm, and as a consequence focusses predominantly on the negative effects of living in poor neighbourhoods. Our study serves as an inspiration for both research and policy focused on the spatial transmission and segregation of affluence. The positive effect of growing up in an affluent neighbourhood is not a serendipitous turn of fate; urban segregation is an outcome of opportunity hoarding processes by those with the means to do so, even if people do not expect the macro level outcomes of their decisions [as in, for example, the Schelling ethnic segregation models: 59], and the overwhelming majority of households are subjected to the whims of landlords and developers controlling the housing market. By studying the effects of living in both affluent and poor environments, we have painted a fuller picture in which urban segregation is not just driven by the sociospatial transmission of deprivation, but also by most resources being concentrated in affluent neighbourhoods.

## Supporting information

**S1 Appendix.**
(DOCX)

## Acknowledgments

The authors would like to acknowledge and thank for Heleen J. Janssen's contributions during the early stages of the work on this manuscript.

## Author Contributions

**Conceptualization:** Agata A. Troost.

**Data curation:** Agata A. Troost.

**Formal analysis:** Agata A. Troost.

**Funding acquisition:** Maarten van Ham.

**Investigation:** Agata A. Troost.

**Methodology:** Agata A. Troost, Maarten van Ham.

**Project administration:** Maarten van Ham.

**Supervision:** Maarten van Ham, David J. Manley.

**Writing – original draft:** Agata A. Troost.

**Writing – review & editing:** Agata A. Troost, Maarten van Ham, David J. Manley.

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
