## [Decision Letter · Decision Letter 0]

10 May 2022

PONE-D-22-09411Neighbourhood effects on educational attainment. What matters more: exposure to poverty or exposure to affluence?PLOS ONE

Dear Dr. Troost,

Thank you for submitting your manuscript to PLOS ONE. After careful consideration, we feel that it has merit but does not fully meet PLOS ONE’s publication criteria as it currently stands. Therefore, we invite you to submit a revised version of the manuscript that addresses the points raised during the review process.

Please ensure you address all the comments raised by the reviewers, both on the methods and analysis presented in the paper, as well as in the framing of arguments, literature, and discussion of the work.

We look forward to receiving your revised manuscript.

Kind regards,

Federico Botta

Academic Editor

PLOS ONE

Journal Requirements:

4. We note you have included a table to which you do not refer in the text of your manuscript. Please ensure that you refer to Table 1 in your text; if accepted, production will need this reference to link the reader to the Table.

Reviewers' comments:

Reviewer's Responses to Questions

**Comments to the Author**

1. Is the manuscript technically sound, and do the data support the conclusions?

Reviewer #1: Partly

Reviewer #2: Partly

2. Has the statistical analysis been performed appropriately and rigorously? 

Reviewer #1: Yes

Reviewer #2: Yes

3. Have the authors made all data underlying the findings in their manuscript fully available?

Reviewer #1: No

Reviewer #2: No

4. Is the manuscript presented in an intelligible fashion and written in standard English?

Reviewer #1: Yes

Reviewer #2: Yes

5. Review Comments to the Author

Reviewer #1: Contribution: The paper compares the effects of exposure to neighbourhood affluence and poverty on educational attainment using Netherlands data.

General comments: The paper is clear on its goal and relevance. I would recommend the authors to check the PLOS ONE template. If I am not mistaken, there is another standard format.

Specific comments (Everything that I had to read again is included here, even the obvious parts.):

[1] The introduction and the literature review are really clear on the relevance of the study, but I would recommend making them shorter. It felt too long, and too repetitive sometimes.

[2] From "This gap is striking as patterns of socio-economic segregation in cities are largely driven by the residential choices of affluent households.", which gap is striking? Are the patterns driven by residential choices or papers/data can not isolate the residential choice factor? I would also add a reference here.

[3] "better educated neighbours", not sure better is the right word here.

[4] On Page 19, what this expression "(see for instance 19)" is referring? citation?

[5] On the end of the introduction, I would recommend adding a summary of the results. The authors for instance say "we test if the exposure to the neighborhood context ...", why dont you add what you found? I do find relevant to restate the contributions on the end of the introduction.

[6] From "This paper addresses the issue of the poverty paradigm in the literature specifically paying attention to the other side of the inequality coin: spatially concentrated affluence. " - would you say that there are only two sides? is it a coin?

[7] What is your point here? : "The empirical nature of such

papers, and the strict paper structure characteristic for the middle-range social studies, usually

does not allow for extensive theoretical commentary about inequality. Nevertheless, the

concepts used in these papers are based on a variety of competing approaches to class, status

and inequality (for an early overview see 30), even if these inspirations are not immediately

visible." - It felt unnecessary to me.

[8] "won't show the same kind of assertiveness" - I would make this part more formal.

[9] What are the theoretical assumptions? "The scale of spatial research should be chosen

according to the theoretical assumptions of the study (50), and in our case we focus on

relatively small-scale, social-interactive neighbourhood effects which would happen in

neighbourhoods of about 200 households."

[10] Can you position Table 1 in the same page?

[11] What are the theoretical reasons? "We

argued that there are theoretical reasons to believe that exposure to affluence might actually be

more important as a predictor of educational attainment than exposure to poverty"

[12] From "The main outcome of this paper is that the contextual effect of neighbourhood affluence

is stronger than the effect of neighbourhood poverty. This confirms that affluence plays a

crucial role in the spatial reproduction of inequalities." -> Confirms the educational attainment or the reproduction of inequalities? Which inequalities?

[13] The images are not in a good resolution.

Reviewer #2: # Neighbourhood effects on educational attainment. What matters more: exposure to poverty or exposure to affluence?

# Summary

This paper first argues that the existing literature on neighborhood effects on individual outcomes misses a large, longstanding theoretical concept: concentration of affluence. To do this, they first present a theoretical argument grounded in sociological and political theory. They then present results from an empirical study investigating the difference between measures of concentrated affluence and concentrated poverty on individual educational attainment using a series of linear regression models. The empirical results generally support the paper’s hypothesis in the context of the Netherlands.

Overall, i enjoyed this paper and think the argument the authors are making is sound, and an important contribution to the conversation around quantiative studies of individual attainment, poverty, and spatial influence. The modeling, while simple, is a totally reasonable approach and the results are largely clear. Although the theoretical discussion could be re-structured, I really enjoyed it and applaud the authors for bringing this perspective to the literature. However, there are a number of things the authors could do to improve the paper further, particularly in the methods and analysis, that I would like to see. While most of my suggests are aimed at improving the clarity and rigor of the paper, not changing the paper entirely, I still recommend a major revision for this work.

## Strengths

### Theoretical framing

I overall enjoyed the theoretical framing of the paper. It is absolutely true that the literature over-focuses on the opportunity hoarding and individual attributes approaches. I also like the small insights into the literature nestled throughout the paper, such as the argument that using categorical income measures makes researchers more likely to focus on poverty.

### Analysis

I thought the k-nearest-neighbors approach was clever and was a good way of addressing heterogeneity in your dataset. The creation of the poverty and affluence variables was also very sensible. Results are straightforward and clear.

### Suggestions: intro, related work, background

- The theoretical background section is actually an argument, rather than a neutral background. The authors should make this more clear by, for example, adding a sentence or two in the first paragraph of the theoretical section saying “In this section, we argue that the effects of concentrated affluence

- It’s unclear to the reader how precisely the theoretical background fits in with the rest of the paper until the reader arrives at the “Current Study” section. The authors could make this more clear in the theoretical background.

- Although the authors reference many spatial inequality studies, they could cite more and be in more in-depth conversation with their approaches for the reader’s benefit.

- To address these comments, I think the theoretical and background section could be restructured to be more effective and clear. it is a combination of a critique of the existing literature and an overview of the theoretical processes the existing quantiative and spatial literature rests on. I suggest that the authors split these two goals apart into two sections. The first section could discuss explicitly recent quantitative work in the field in a more neutral manner. The second could use this grounding to critique the existing field while introducing the theoretical concepts that the paper leans on heavily.

### Suggestions: methods and analysis

- Some of the spatial variable creation could be a little unclear to some readers. My understanding is that each individual in your dataset has a grid square assigned to them as their “home” location for each time period. Within each time period, for each subject, the authors gather data on the 200 nearest households from that year, and use this household data to compute the affluence and poverty metrics. If this is the case, the authors should make this a bit clearer. As it stands, it’s a bit unclear the relationship between the grid cells and households, and if KNN was run with grid cells or historical household data.

- If spatial context is important for social interaction, etc., I’d like to see some measure of the average distance to different affluent or poor households included in the analysis. One example metric could be the mean geographic distance from grid cell to affluent households. These results may be more nuanced and reveal a bit more in terms of mechanism, which the authors are clearly interested in.

- I would also like to see much more descriptive and analytical content aimed at the measures the authors calculated. The KNN measure over years of subjects’ lives is very interesting! I want to know how this measure changes over time for different individuals. What mediates those changes? Similarly, like chetty’s study, the authors could look at *changes* in these metrics to investigate their true impact.

- Similarly, I would be interested in a model that just looks at these metrics at year 8 (2003). Are the effects and the rho similar? If so, the authors should include the framing of measuring early childhood environment to predict later outcomes.

- I think the affluent and poverty variables may be correlated, because they are calculated as percentages of the same total. I’d like to see the authors address this either through diagnostic tests or in the text of the paper.

- The authors do not present a “null” model without their spatial poverty and affluence concentration variables. I would like to see how much extra variance these variables contribute to the model. Without this measure of comparison, it’s hard to find the results very meaningful.

- I found the argument for not doing a more complicated model a little weak, and confusing at times:

- Does the explanation of “nested” individuals mean that there are many subjects who are “alone” in their cluster? If so, doesn’t this follow from the methods? Why would two individuals share the same KNN, unless they lived in the same grid cell? The authors should make this more clear.

- The authors could define what they mean by “neighborhood” here—is it the statistical / political area, or the KNN you defined above?

- I agree that a traditional multilevel model at the “neighborhood” level would likely be too complex here. But I do think the authors could include some fixed effects for general region if people are highly spatial distributed. For example, a fixed effect added for each “year x region” combination may be important. As it stands now, there are no general covariates controlling for other, unmeasured attributes of a region.

### Other Suggestions

- you only mention chetty’s study in your conclusion, but they focused heavily on the impact of affluence on educational attainment. address that!

- Although the paper is written well, there are, generally, some non-english-isms scattered throughout the paper which have made it a little hard to read as a native english speaker. The authors might consider getting a native english speaker to edit the work before publication for clarity.

- Figure 1 should be re-labeled and re-rendered—it’s very fuzzy in the copy I received and having the titles and axes in more plain english would be helpful.

- In the conclusion, the authors cite schelling differently than in the rest of the paper?

6. PLOS authors have the option to publish the peer review history of their article (what does this mean?). If published, this will include your full peer review and any attached files.

Reviewer #1: No

Reviewer #2: No

---

## [Author Response · Author response to Decision Letter 0]

2 Sep 2022

(See the submitted file for better formatting)

Dear Dr. Botta, 

Thank you very much for the opportunity to revise and resubmit our manuscript entitled “Neighbourhood effects on educational attainment. What matters more: exposure to poverty or exposure to affluence?” (PONE-D-22-09411). And thank you for considering the paper for publication. 

Based on the reviewer comments we have thoroughly revised the paper. We have restructured the paper, shortened some parts and expanded others. In this process we had to make choices, such as in the theory section as reviewer 1 asked to shorten it somewhat while reviewer 2 asked to expand and restructure some parts. We have also revised and shortened the introduction, as well as added extra explanation to the methods and analysis section and clarified some fragments pointed out by the reviewers. Overall, we feel that the reviewer comments have helped us to substantially sharpen the paper and we are grateful for their time. Please see the attached detailed replies to the reviewers’ comments for further explanation of the changes we made. 

When it comes to the journal requirements regarding ethics, we work with confidential Statistics Netherlands (Centraal Bureau voor Statistiek, CBS) data. CBS is legally responsible for consent and our project has been approved by them (to get access to the data). All statistical output based on the data is checked by CBS employees to make sure it respects the privacy of the subjects. CBS is bound by the European General Data Protection Regulation (GDPR). In addition, CBS adheres to the privacy stipulations in the Statistics Netherlands Act, the European Statistics Code of Practice, and its own Code of conduct (Dutch only). The links to these documents and more explanation can be found here: https://www.cbs.nl/en-gb/about-us/organisation/privacy. Because of these privacy rules, we cannot share the data underlying our results. We have now added this clarification to the manuscript text in the Methods section (p. 14). We have also added a reference to Table 1 in the text. 

We hope the revised manuscript meets your expectations.

Best wishes, 

Agata Troost, Maarten van Ham and David Manley 

Reviewer 1 

COMMENT:

Contribution: The paper compares the effects of exposure to neighbourhood affluence and poverty on educational attainment using Netherlands data.

General comments: The paper is clear on its goal and relevance. I would recommend the authors to check the PLOS ONE template. If I am not mistaken, there is another standard format.

RESPONSE:

Thank you for reviewing our paper! We have now consulted the template and submission guidelines, and adjusted our paper by removing footnotes and changing the headings. 

COMMENT:

Specific comments (Everything that I had to read again is included here, even the obvious parts.):

[1] The introduction and the literature review are really clear on the relevance of the study, but I would recommend making them shorter. It felt too long, and too repetitive sometimes.

RESPONSE:

We have edited the introduction and theory section to shorten it where possible. 

COMMENT:

[2] From "This gap is striking as patterns of socio-economic segregation in cities are largely driven by the residential choices of affluent households.", which gap is striking? Are the patterns driven by residential choices or papers/data can not isolate the residential choice factor? I would also add a reference here.

RESPONSE:

We have reformulated the sentence to make it clearer: “The lack of literature on concentrated affluence is all the more striking given the influential position of affluent households: the choices of the wealthy largely shape patterns of socio-economic segregation in cities, as higher income households can use their resources to select the best residential locations in a city” (p. 3), and added a reference to Troost et al. (2021). 

COMMENT:

[3] "better educated neighbours", not sure better is the right word here.

RESPONSE:

We changed “better” to “higher”. 

COMMENT:

[4] On Page 19, what this expression "(see for instance 19)" is referring? citation?

RESPONSE:

The full sentence (on page 5 in our version) is “Following earlier studies (see for instance 20), we test if the exposure to the neighbourhood context (both affluence and poverty) is different for children with different parental levels of education”. “20” is indeed referring to the cited study by Sykes and Kuyper, who studied neighbourhood effects on children influenced by parental education. We reformulated the sentence to: “Following earlier studies (20)…”, which is more readable while using the Vancouver citation style. 

COMMENT:

[5] On the end of the introduction, I would recommend adding a summary of the results. The authors for instance say "we test if the exposure to the neighborhood context ...", why dont you add what you found? I do find relevant to restate the contributions on the end of the introduction.

RESPONSE:

We have now briefly described our results at the end of introduction (p. 5). 

COMMENT:

[6] From "This paper addresses the issue of the poverty paradigm in the literature specifically paying attention to the other side of the inequality coin: spatially concentrated affluence. " - would you say that there are only two sides? is it a coin?

RESPONSE: We have rewritten the text to remove the unintentional ambiguity in the metaphor: “This paper addresses the issue of the poverty paradigm in the literature by specifically paying attention to spatially concentrated affluence” (p. 5). 

COMMENT:

[7] What is your point here? : "The empirical nature of such papers, and the strict paper structure characteristic for the middle-range social studies, usually

does not allow for extensive theoretical commentary about inequality. Nevertheless, the

concepts used in these papers are based on a variety of competing approaches to class, status

and inequality (for an early overview see 30), even if these inspirations are not immediately

visible." - It felt unnecessary to me.

RESPONSE:

Our intention was to highlight that the influence of spatially concentrated affluence has, partially, been neglected because of the research habits in the field. Because there was not always enough space or focus for the theoretical implications of approaches and theories used, it was easier for studies to remain in the “poverty paradigm” and overlook the crucial role of concentrated affluence. We hope that after the restructuring of the theory section this argument is clearer. 

COMMENT:

[8] "won't show the same kind of assertiveness" - I would make this part more formal.

RESPONSE: 

We have replaced it with the clarification “are unable to mobilise the same degree of social and cultural capital” (p. 10). 

COMMENT:

[9] What are the theoretical assumptions? "The scale of spatial research should be chosen

according to the theoretical assumptions of the study (50), and in our case we focus on

relatively small-scale, social-interactive neighbourhood effects which would happen in

neighbourhoods of about 200 households."

RESPONSE:

We have rewritten this section to make it clearer: “The scale of spatial research should be chosen according to the theoretical assumptions of the study (51), and in our case we focus on relatively small-scale, social-interactive neighbourhood effects which would happen in neighbourhoods of about 200 households. This size should reflect a social space where people are likely to interact with each other, which, according to our assumptions, leads to acquiring skills and resources relevant for an individual’s educational attainment” (p. 16). 

COMMENT:

[10] Can you position Table 1 in the same page?

RESPONSE:

The table is now moved to the next page so it doesn’t break. 

COMMENT:

[11] What are the theoretical reasons? "We argued that there are theoretical reasons to believe that exposure to affluence might actually be more important as a predictor of educational attainment than exposure to poverty"

RESPONSE:

To ensure the theoretical reasons are clearer we have summarised them explicitly (see p.22): “We argued that there are theoretical reasons to believe that exposure to affluence may actually be more important as a predictor of educational attainment than exposure to poverty, because of the crucial influence of interacting with higher educated people on one’s resources, skills and educational aspirations; and, in the Dutch context, because of the lack of extreme concentrated poverty”. 

COMMENT:

[12] From "The main outcome of this paper is that the contextual effect of neighbourhood affluence

is stronger than the effect of neighbourhood poverty. This confirms that affluence plays a

crucial role in the spatial reproduction of inequalities." -> Confirms the educational attainment or the reproduction of inequalities? Which inequalities?

RESPONSE:

We rewrote the beginning of that paragraph to achieve more clarity and a better flow: “Most importantly, our results highlight how spatially concentrated affluence contributes to the reproduction of socioeconomic inequalities, as the effect of neighbourhood affluence on educational attainment is stronger than that of neighbourhood poverty” (p. 23).

COMMENT:

[13] The images are not in a good resolution.

RESPONSE: 

We have now attached a figure in a high quality tiff format. However, we understand that the figure at the end of the preview pdf will always be blurry unless downloaded separately. 

Reviewer 2

COMMENT:

# Neighbourhood effects on educational attainment. What matters more: exposure to poverty or exposure to affluence?

# Summary

This paper first argues that the existing literature on neighborhood effects on individual outcomes misses a large, longstanding theoretical concept: concentration of affluence. To do this, they first present a theoretical argument grounded in sociological and political theory. They then present results from an empirical study investigating the difference between measures of concentrated affluence and concentrated poverty on individual educational attainment using a series of linear regression models. The empirical results generally support the paper’s hypothesis in the context of the Netherlands.

Overall, i enjoyed this paper and think the argument the authors are making is sound, and an important contribution to the conversation around quantitative studies of individual attainment, poverty, and spatial influence. The modeling, while simple, is a totally reasonable approach and the results are largely clear. Although the theoretical discussion could be re-structured, I really enjoyed it and applaud the authors for bringing this perspective to the literature. However, there are a number of things the authors could do to improve the paper further, particularly in the methods and analysis, that I would like to see. While most of my suggests are aimed at improving the clarity and rigor of the paper, not changing the paper entirely, I still recommend a major revision for this work.

## Strengths

### Theoretical framing

I overall enjoyed the theoretical framing of the paper. It is absolutely true that the literature over-focuses on the opportunity hoarding and individual attributes approaches. I also like the small insights into the literature nestled throughout the paper, such as the argument that using categorical income measures makes researchers more likely to focus on poverty.

### Analysis

I thought the k-nearest-neighbors approach was clever and was a good way of addressing heterogeneity in your dataset. The creation of the poverty and affluence variables was also very sensible. Results are straightforward and clear.

RESPONSE:

Thank you for the appreciation and the helpful summary of the paper!

COMMENT:

### Suggestions: intro, related work, background

- The theoretical background section is actually an argument, rather than a neutral background. The authors should make this more clear by, for example, adding a sentence or two in the first paragraph of the theoretical section saying “In this section, we argue that the effects of concentrated affluence

- It’s unclear to the reader how precisely the theoretical background fits in with the rest of the paper until the reader arrives at the “Current Study” section. The authors could make this more clear in the theoretical background.

- Although the authors reference many spatial inequality studies, they could cite more and be in more in-depth conversation with their approaches for the reader’s benefit.

- To address these comments, I think the theoretical and background section could be restructured to be more effective and clear. it is a combination of a critique of the existing literature and an overview of the theoretical processes the existing quantitative and spatial literature rests on. I suggest that the authors split these two goals apart into two sections. The first section could discuss explicitly recent quantitative work in the field in a more neutral manner. The second could use this grounding to critique the existing field while introducing the theoretical concepts that the paper leans on heavily.

RESPONSE:

We have now restructured the theory section and start with the overview of empirical studies, which then leads to the theoretical reasons for including affluence in the study, which in turn leads to the “Current study” section. We also added some new references, like Imbroscio and Custers & Engbersen, to add further context to this section. In addition, we used some more phrases implying we are constructing an argument. 

COMMENT:

### Suggestions: methods and analysis

- Some of the spatial variable creation could be a little unclear to some readers. My understanding is that each individual in your dataset has a grid square assigned to them as their “home” location for each time period. Within each time period, for each subject, the authors gather data on the 200 nearest households from that year, and use this household data to compute the affluence and poverty metrics. If this is the case, the authors should make this a bit clearer. As it stands, it’s a bit unclear the relationship between the grid cells and households, and if KNN was run with grid cells or historical household data.

RESPONSE:

Yes! This is how we approached the neighborhoods. We agree that the text could be clearer and we have adjusted it to address this (eg. that the measures change for each year within the analysis). 

COMMENT:

- If spatial context is important for social interaction, etc., I’d like to see some measure of the average distance to different affluent or poor households included in the analysis. One example metric could be the mean geographic distance from grid cell to affluent households. These results may be more nuanced and reveal a bit more in terms of mechanism, which the authors are clearly interested in.

RESPONSE:

Thank you for this comment. We had given it a lot of thought but it is not straightforward to make it operational. We can only measure difference from a cell to a cell, and a similar variable would most likely just show that a bigger geographic distance is required to reach enough affluent/poor neighbours in less densely populated areas, which is already controlled for with the “urbanisation” variable. In addition, a mean variable would just “flatten out” the average necessary distance, with it being shorter for poor households having to reach other poor (because of poor households being more prevalent in that area), etc. It could be interesting to compare similar measures for different cities or districts, but they would most likely correspond closely to the poverty/affluence ratios of bespoke neighbourhoods we already have. 

COMMENT:

- I would also like to see much more descriptive and analytical content aimed at the measures the authors calculated. The KNN measure over years of subjects’ lives is very interesting! I want to know how this measure changes over time for different individuals. What mediates those changes? Similarly, like chetty’s study, the authors could look at *changes* in these metrics to investigate their true impact.

RESPONSE:

We have studied the influence of subjects’ neighbourhood trajectories / individual neighbourhood histories sequences in our previous study, and considered adding the data on them to this paper as an extra context (percentages of individuals with constant low poverty neighbourhood histories, constant low affluence, constant medium poverty etc…). However, because of word count limitations and no clear theoretical relevance of these sequences to the rest of the paper, we have decided to omit this extra information in the current paper. Our other paper discusses in depth the differences between different longitudinal measurements of the bespoke neighbourhoods. The statistical models we are using in this paper were chosen as the clearest and most relevant ones for the affluence/poverty comparison. 

COMMENT:

- Similarly, I would be interested in a model that just looks at these metrics at year 8 (2003). Are the effects and the rho similar? If so, the authors should include the framing of measuring early childhood environment to predict later outcomes.

RESPONSE: We have looked at the models and while the effects of one year compared to multiple years are similar, we believe that including longer measurement periods provides a better reflection of the underlying processes we are exploring and the accumulation of live deposits. We already look at both early childhood and adolescence, in addition to the total childhood years, predicting our main outcome of interest, educational attainment. 

COMMENT:

- I think the affluent and poverty variables may be correlated, because they are calculated as percentages of the same total. I’d like to see the authors address this either through diagnostic tests or in the text of the paper.

RESPONSE:

The correlation between the variables is -.45, therefore the correlation is not problematic for the models. We have now added this information to the text on page 16 (“The variables are not correlated (-.45)”). 

COMMENT:

- The authors do not present a “null” model without their spatial poverty and affluence concentration variables. I would like to see how much extra variance these variables contribute to the model. Without this measure of comparison, it’s hard to find the results very meaningful.

RESPONSE:

The spatial variables contribute around 3% difference in R-squared (from around 15% with no spatial variables, through 16% with only the urbanisation control, to 18% with all spatial variables). While it doesn’t seem to be a big difference, it is the type of difference that can be expected from similar variables in sociological models. Additionally, including the spatial variables diminishes the effects of other variables in the model, such as family income, which means the spatial variables contribute to the underlying causal structures. 

COMMENT:

- I found the argument for not doing a more complicated model a little weak, and confusing at times:

- Does the explanation of “nested” individuals mean that there are many subjects who are “alone” in their cluster? If so, doesn’t this follow from the methods? Why would two individuals share the same KNN, unless they lived in the same grid cell? The authors should make this more clear.

RESPONSE:

We have now rewritten the section and added more explanation to make it clearer (on p. 18): “Given the nested structure of our data, the use of multilevel modelling appears logical. However, there are two reasons why we have not used this type of models. Firstly, individuals are nested in neighbourhoods which can change every year. Therefore, the complex hierarchical structure inhibits model convergence. This is further exacerbated by the second reason, whereby there is no strict hierarchy because of the multiple membership of individuals in the bespoke neighbourhoods (the neighbourhoods are overlapping with each other). Furthermore, because of bespoke neighbourhoods which are constructed for each individual every year, and only including people born in 1995 in the sample, a large number of individuals are “nested” alone or with just one other person in their neighbourhood (73,367; 49%), which is another obstacle to estimating a hierarchical fixed effects structure.”

COMMENT:

- The authors could define what they mean by “neighborhood” here—is it the statistical / political area, or the KNN you defined above?

RESPONSE:

Unfortunately we do not know to which line this comment refers to, specifically, but in general we refer to the 200-nearest neighbouring households, bespoke neighbourhoods we created. In revising our paper we have paid attention to this issue to ensure that the meaning of neighborhood is clear. 

COMMENT:

- I agree that a traditional multilevel model at the “neighborhood” level would likely be too complex here. But I do think the authors could include some fixed effects for general region if people are highly spatial distributed. For example, a fixed effect added for each “year x region” combination may be important. As it stands now, there are no general covariates controlling for other, unmeasured attributes of a region.

RESPONSE:

We have an urbanisation control variable, but none about the region. However, the Netherlands’ regions do not significantly differ when it comes to economic performance, education quality, and other relevant variables we can think of. There are differences between big cities and more rural regions, but these are captured by the urbanisation control variable. The quality of eg. high schools is the same in Amsterdam, Nijmegen or Maastricht. Regions which are “less economically developed” are also less urbanised, eg. The province of Groningen consists mostly of farming fields, while provinces in the Randstad, such as Utrecht and South Holland, are highly urbanised. This has to do with the Netherlands being a very small, densely populated country in general. Additionally, given the number of years in the analysis, creating so many interaction variables with region per year would not be workable. 

COMMENT:

### Other Suggestions

- you only mention chetty’s study in your conclusion, but they focused heavily on the impact of affluence on educational attainment. address that!

RESPONSE:

We have now added a reference to Chetty earlier in the paper. However, the “better neighbourhoods” in Chetty et al. are neighbourhoods with low(er) poverty rather than high affluence, so we could not identify in it an operationalisation of affluence that would be of high relevance to our paper. 

COMMENT:

- Although the paper is written well, there are, generally, some non-english-isms scattered throughout the paper which have made it a little hard to read as a native english speaker. The authors might consider getting a native english speaker to edit the work before publication for clarity.

RESPONSE:

Thank you, a native English speaker has checked the paper. 

COMMENT:

- Figure 1 should be re-labeled and re-rendered—it’s very fuzzy in the copy I received and having the titles and axes in more plain english would be helpful.

RESPONSE:

We have now adjusted the labels and made sure to attach a figure in a high quality tiff format to the submission. However, the figure at the end of the preview pdf might still be blurry because of compression. 

COMMENT:

- In the conclusion, the authors cite schelling differently than in the rest of the paper

RESPONSE:

Thank you! We have now corrected the reference.

---

## [Decision Letter · Decision Letter 1]

17 Oct 2022

PONE-D-22-09411R1Neighbourhood effects on educational attainment. What matters more: exposure to poverty or exposure to affluence?PLOS ONE

Dear Dr. Troost,

Thank you for submitting your manuscript to PLOS ONE. After careful consideration, we feel that it has merit but does not fully meet PLOS ONE’s publication criteria as it currently stands. Therefore, we invite you to submit a revised version of the manuscript that addresses the points raised during the review process.

Whist both reviewers acknowledge an  improvement in the submission, there are still a number of significant outstanding issues that need to be addressed raised by both reviewers. I encourage the authors to address all remaining comments in the revised submission.

We look forward to receiving your revised manuscript.

Kind regards,

Federico Botta

Academic Editor

PLOS ONE

Reviewers' comments:

Reviewer's Responses to Questions

**Comments to the Author**

1. If the authors have adequately addressed your comments raised in a previous round of review and you feel that this manuscript is now acceptable for publication, you may indicate that here to bypass the “Comments to the Author” section, enter your conflict of interest statement in the “Confidential to Editor” section, and submit your "Accept" recommendation.

Reviewer #1: (No Response)

Reviewer #2: (No Response)

2. Is the manuscript technically sound, and do the data support the conclusions?

Reviewer #1: Partly

Reviewer #2: Yes

3. Has the statistical analysis been performed appropriately and rigorously? 

Reviewer #1: Yes

Reviewer #2: Yes

4. Have the authors made all data underlying the findings in their manuscript fully available?

Reviewer #1: No

Reviewer #2: No

5. Is the manuscript presented in an intelligible fashion and written in standard English?

Reviewer #1: No

Reviewer #2: Yes

6. Review Comments to the Author

Reviewer #1: The paper shows that the neighborhood affluence has a stronger effect on educational attainment than neighborhood poverty in the Netherlands.

Even though the paper was improved, there are still inadequate sentences such as "These results highlight the need for more studies on the effects of concentrated affluence, and they can inspire policies focused on the segregation of richer households." - What do you mean "can inspire policies focused on the segregation"? Is segregation something positive in this context? Why?

Several informal expressions such as "Perhaps","In doing so", "This is what we had in mind with the current paper", "according to our assumptions

Excessive use of "" in the paper

Multiple places without references using very strong statements such as "there are theoretical reasons to believe that exposure to affluence may actually be more important as a predictor of educational attainment than exposure to poverty","As already discussed, poverty is often associated with crime and isolation of minority groups."

The limitations of the work are not in line with the following statement: "we have painted a fuller picture in which the spatial transmission of poverty is not an isolated problem, but one reinforced by most resources being concentrated somewhere else."

I am not sure whether it can be also concluded that "Our results support our initial idea that it is often the lack of resources in poor and middle income neighborhoods that is the problem, not the theorized negative effects of poverty itself."

As the subject of the paper is inequality, I would recommend the authors to tune down the statements (especially the ones that imply "cause and consequence") and use more formal and well accepted jargon.

Reviewer #2: - Thanks to the authors for making these significant adjustments to the paper. I think the structure and presentation are clearer, and many of comments have been addressed. Kudos!

In general I recommend it for publication, but I would strongly encourage the authors to include these final changes in the submitted version for the reader. I think adding these will help you convince more quantitatively minded readers of your argument.

- I could be missing it, but if you do study neighborhood trajectories from the KNN approach in a prior study like you mention in your rebuttal, please reference it when discussing the KNN method or results so that a reader can find that work fairly easily. It is a logical train of thought to want to learn more about the construct you’ve made.

- I am still a little concerned about the affluent and poverty variables. While not quite a composition, it is close to one as the two variables before being averaged across years are constrained. The correlation is also strongly *negative*, not “not correlated” at -.45. I would just mention this and add an argument why you feel there is no need to transform the variables (e.g. see https://link.springer.com/chapter/10.1007/978-3-642-36809-7_5), or include the VIF for your models to convince the reader that there is not a multicollinearity problem.

- I would also include the reasoning you gave me for not including the 'average distance' to different affluent or poor households, mainly because I do not find your answer totally satisfying. You argue that 200 households is a good proxy for social interaction, but the likelihood of interaction with the nearest 200 households for a rural community vs a dense urban one are wildly different in my view. I don't think that controlling for urban density ("urbanity") is enough to capture this. I'd like to see either some text addressing this specifically on e.g. page 16, or some statistics showing me that the urbanization variable highly correlates with a household's avg. distance to poor and affluent households. If it doesn't, I think it would show that there is variation your analysis isn't capturing. Even if that's the case, you can show that and then mention why you don't operationalize it. As it stands it's still a question I have reading the paper.

- Regarding the “null” model, thanks to the authors for their explanation. However, I would like the explanation you gave me in the review to be present in the text.

7. PLOS authors have the option to publish the peer review history of their article (what does this mean?). If published, this will include your full peer review and any attached files.

Reviewer #1: No

Reviewer #2: No

---

## [Author Response · Author response to Decision Letter 1]

31 Dec 2022

(see also the doc file submitted earlier)

Dear Dr. Botta, 

Thank you very much for the opportunity to revise our manuscript entitled “Neighbourhood effects on educational attainment. What matters more: exposure to poverty or exposure to affluence?” (PONE-D-22-09411R1). And thank you for considering the paper for publication. 

We have addressed the reviewers’ comments individually, in the text below. In general, according to the comments of Reviewer 1 we made the text more specific and formal. Following the comments of Reviewer 2, we have added a control variable and we have clarified a number of methodological assumptions and conditions in the text. We feel that the reviewer comments have helped us to further improve the paper and we are grateful for their time.

In one of the comments Reviewer 2 asks us to add a specific reference to our earlier work in the text. In order for the reviewers to still be able to anonymously assess the manuscript, we have replaced this reference in text with “[REDACTED]”. The reference should be: 

[Redacted for anonymity - please see the comments to the journal office / the doc file]

We hope the revised manuscript meets your expectations.

Best wishes, 

[Redacted for anonymity - please see the comments to the journal office / the doc file]

REVIEWER 1

COMMENT:

Reviewer #1: The paper shows that the neighborhood affluence has a stronger effect on educational attainment than neighborhood poverty in the Netherlands. 

Even though the paper was improved, there are still inadequate sentences such as "These results highlight the need for more studies on the effects of concentrated affluence, and they can inspire policies focused on the segregation of richer households." - What do you mean "can inspire policies focused on the segregation"? Is segregation something positive in this context? Why?

RESPONSE:

We have now rephrased that sentence as: “These results highlight the need for more studies on the effects of concentrated affluence, and they can inspire anti-segregation policies focused on the concentration of rich households” (p. 2). 

COMMENT:

Several informal expressions such as "Perhaps","In doing so", "This is what we had in mind with the current paper", "according to our assumptions

Excessive use of "" in the paper

RESPONSE:

We have made the language more formal and reduced the usage of quotation marks where possible. 

COMMENT:

Multiple places without references using very strong statements such as "there are theoretical reasons to believe that exposure to affluence may actually be more important as a predictor of educational attainment than exposure to poverty","As already discussed, poverty is often associated with crime and isolation of minority groups."

RESPONSE:

For the second cited statement, we have now added references to support it (p. 8):

DeLuca S, Duncan GJ, Keels M, Mendenhall R. The notable and the null: Using mixed methods to understand the diverse impacts of residential mobility programs. In: Neighbourhood effects research: New perspectives. Springer; 2012. p. 195–223. 

Sharkey P. Uneasy peace: The great crime decline, the renewal of city life, and the next war on violence. WW Norton & Company; 2018.

The first statement, however, is in the beginning of the conclusion section and summarises our theoretical background section argument, based on multiple studies. We therefore do not think adding one or two references would make sense here, or even referring to all the studies we referenced in theoretical background to make our argument, as the sentence refers to the argument we formulated ourselves. Its function as a summary is clearer while looking at the full sentence (p. 23): “We argued that there are theoretical reasons to believe that exposure to affluence may actually be more important as a predictor of educational attainment than exposure to poverty, because of the crucial influence of interacting with higher educated people on one’s resources, skills and educational aspirations; and, in the Dutch context, because of the lack of extreme concentrated poverty.”

COMMENT:

The limitations of the work are not in line with the following statement: "we have painted a fuller picture in which the spatial transmission of poverty is not an isolated problem, but one reinforced by most resources being concentrated somewhere else."

RESPONSE:

We have reformulated the sentence as “By studying the effects of living in both affluent and poor environments, we have painted a fuller picture in which urban segregation is not just driven by the sociospatial transmission of deprivation, but also by most resources being concentrated in affluent neighbourhoods” (p. 25). 

COMMENT:

I am not sure whether it can be also concluded that "Our results support our initial idea that it is often the lack of resources in poor and middle income neighborhoods that is the problem, not the theorized negative effects of poverty itself."

RESPONSE:

We have now added more details to make the sentence clearer: “Our results, specifically the effect of spatially concentrated affluence being stronger than that of poverty, support our initial idea that it is often the lack of resources – the cultural and economic capital of richer neighbours - in poor and middle income neighbourhoods that is the problem, not the theorised negative effects of poverty itself” (p. 24).

COMMENT:

As the subject of the paper is inequality, I would recommend the authors to tune down the statements (especially the ones that imply "cause and consequence") and use more formal and well accepted jargon.

RESPONSE:

We have edited the statements to be more nuanced (as also detailed in the previous comments) and made the language of the paper more formal. 

REVIEWER 2

COMMENT:

Reviewer #2: - Thanks to the authors for making these significant adjustments to the paper. I think the structure and presentation are clearer, and many of comments have been addressed. Kudos!

In general I recommend it for publication, but I would strongly encourage the authors to include these final changes in the submitted version for the reader. I think adding these will help you convince more quantitatively minded readers of your argument.

RESPONSE:

Thank you for the kind words and useful comments!

COMMENT:

- I could be missing it, but if you do study neighborhood trajectories from the KNN approach in a prior study like you mention in your rebuttal, please reference it when discussing the KNN method or results so that a reader can find that work fairly easily. It is a logical train of thought to want to learn more about the construct you’ve made.

RESPONSE:

As the study we mentioned has been published in the meantime, we have added references to it (p. 16, p. 25). We marked the reference in the manuscript as [REDACTED] and sent the actual reference to the editor, as sharing it with the reviewers in possible future rounds of reviews would mean that the review is not double blind anymore. 

COMMENT:

- I am still a little concerned about the affluent and poverty variables. While not quite a composition, it is close to one as the two variables before being averaged across years are constrained. The correlation is also strongly *negative*, not “not correlated” at -.45. I would just mention this and add an argument why you feel there is no need to transform the variables (e.g. see https://link.springer.com/chapter/10.1007/978-3-642-36809-7_5), or include the VIF for your models to convince the reader that there is not a multicollinearity problem.

RESPONSE:

We have now added this explanation, on p. 19: “VIF values were unproblematic, therefore there are no issues with multicollinearity in the models (see Appendix for exact VIF values)”, and also changed “not correlated” to “weakly correlated”. 

COMMENT:

- I would also include the reasoning you gave me for not including the 'average distance' to different affluent or poor households, mainly because I do not find your answer totally satisfying. You argue that 200 households is a good proxy for social interaction, but the likelihood of interaction with the nearest 200 households for a rural community vs a dense urban one are wildly different in my view. I don't think that controlling for urban density ("urbanity") is enough to capture this. I'd like to see either some text addressing this specifically on e.g. page 16, or some statistics showing me that the urbanization variable highly correlates with a household's avg. distance to poor and affluent households. If it doesn't, I think it would show that there is variation your analysis isn't capturing. Even if that's the case, you can show that and then mention why you don't operationalize it. As it stands it's still a question I have reading the paper.

RESPONSE:

Thank you for this comment. We have discussed the issue of the average distance again. Because the correlation between the average distance (measured by Equipop in kilometres necessary to reach the 200 nearest households) and urbanicity (as we decided to rename the variable more accurately) isn’t strong (-0.37 for both high and low income), we decided to add the average distance as a control variable in our models. We believe it helps to control for density of social interaction at a lower level than the urbanicity variable. Adding the variable did not change the significance of any of the other variables except for being of a Western migrant background, but it did influence the strength of the observed effects, making the influence of spatially concentrated affluence stronger than before, and that of poverty a bit weaker. The N of the dataset for analyses changed to 140,338 because of that, but as this indicated that previously some of the bespoke neighbourhood-based variables included squares with missing information for some years, we believe this is an improvement of the study. 

COMMENT:

- Regarding the “null” model, thanks to the authors for their explanation. However, I would like the explanation you gave me in the review to be present in the text.

RESPONSE:

We have now added the explanation to the text on page 19: “The spatial variables contribute around 3% difference in R-squared – from around 15% in models with no spatial variables (for detailed coefficients, see Appendix), through 16% with only the urbanicity control, to 18% with all spatial variables – which is the type of difference that can be expected from similar variables in sociological models. Additionally, including the spatial variables diminishes the effects of other variables in the model, such as family income, which means the spatial variables contribute to the underlying causal structures.”

---

## [Decision Letter · Decision Letter 2]

6 Feb 2023

Neighbourhood effects on educational attainment. What matters more: exposure to poverty or exposure to affluence?

PONE-D-22-09411R2

Dear Dr. Troost,

We’re pleased to inform you that your manuscript has been judged scientifically suitable for publication and will be formally accepted for publication once it meets all outstanding technical requirements.

Kind regards,

Federico Botta

Academic Editor

PLOS ONE

Additional Editor Comments (optional):

Reviewers' comments:

Reviewer's Responses to Questions

**Comments to the Author**

1. If the authors have adequately addressed your comments raised in a previous round of review and you feel that this manuscript is now acceptable for publication, you may indicate that here to bypass the “Comments to the Author” section, enter your conflict of interest statement in the “Confidential to Editor” section, and submit your "Accept" recommendation.

Reviewer #1: All comments have been addressed

2. Is the manuscript technically sound, and do the data support the conclusions?

Reviewer #1: Yes

3. Has the statistical analysis been performed appropriately and rigorously? 

Reviewer #1: Yes

4. Have the authors made all data underlying the findings in their manuscript fully available?

Reviewer #1: No

5. Is the manuscript presented in an intelligible fashion and written in standard English?

Reviewer #1: Yes

6. Review Comments to the Author

Reviewer #1: No further comments.

The paper was greatly improved from its first version.

Thank you for addressing all my comments.

7. PLOS authors have the option to publish the peer review history of their article (what does this mean?). If published, this will include your full peer review and any attached files.

Reviewer #1: No

---

## [Editor Report · Acceptance letter]

14 Feb 2023

PONE-D-22-09411R2 

Neighbourhood effects on educational attainment. What matters more: exposure to poverty or exposure to affluence? 

Dear Dr. Troost:

I'm pleased to inform you that your manuscript has been deemed suitable for publication in PLOS ONE. Congratulations! Your manuscript is now with our production department. 

Kind regards, 

on behalf of

Dr. Federico Botta 

Academic Editor

PLOS ONE